# Senescent cell turnover slows with age providing an explanation for the Gompertz law

Omer Karin[1,3], Amit Agrawal[1,3], Ziv Porat [2], Valery Krizhanovsky [1]* & Uri Alon [1]*

A causal factor in mammalian aging is the accumulation of senescent cells (SnCs). SnCs cause chronic inflammation, and removing SnCs decelerates aging in mice. Despite their importance, turnover rates of SnCs are unknown, and their connection to aging dynamics is unclear. Here we use longitudinal SnC measurements and induction experiments to show that SnCs turn over rapidly in young mice, with a half-life of days, but slow their own removal rate to a half-life of weeks in old mice. This leads to a critical-slowing-down that generates persistent SnC fluctuations. We further demonstrate that a mathematical model, in which death occurs when fluctuating SnCs cross a threshold, quantitatively recapitulates the Gompertz law of mortality in mice and humans. The model can go beyond SnCs to explain the effects of lifespan-modulating interventions in *Drosophila* and *C. elegans*, including scaling of survival-curves and rapid effects of dietary shifts on mortality.

[1] Department of Molecular Cell Biology, Weizmann Institute of Science, 76100 Rehovot, Israel. [2] Department of Biological Services, Weizmann Institute of Science, 76100 Rehovot, Israel. [3]These authors contributed equally: Omer Karin, Amit Agrawal. *email: valery.krizhanovsky@weizmann.ac.il; uri.alon@weizmann.ac.il

Senescent cells (SnCs) accumulate with age in mice and humans in many tissues[1–7], due in part to DNA damage, damaged telomeres, and oxidative stress[5,8]. These cells, characterized by high levels of p16 and SA-β-Gal[5], enter permanent cell cycle arrest, and secrete a characteristic profile of molecules including pro-inflammatory signals[9] and factors that slow regeneration[9] (Fig. 1a). They have physiological roles in development, cancer prevention, and wound healing[9–11]. However, as organisms age, accumulating levels of SnC cause chronic inflammation and increase the risk of many age-related diseases, including osteoarthritis, neurodegeneration, and atherosclerosis[12–24].

Accumulation of SnCs is known to be causal for aging in mice: continuous targeted elimination of whole-body SnCs increases mean lifespan by 25%, attenuates age-related deterioration of heart, kidney, and fat, delays cancer development[25] and causes improvement in the above-mentioned diseases.

These studies indicate that SnC abundance is an important causal variable in the aging process. Despite their importance, however, the production and removal rates of SnCs are unknown[9,26]. For example, it is unclear whether SnCs passively accumulate or if they are turned over rapidly, and if so, whether their half-life changes with age. Since turnover affects the ability of a system to respond to fluctuations, information about these rates is crucial in order to mathematically test ideas about the possible role of SnCs in the age-dependent variations in morbidity and mortality between individuals.

Here, we address this experimentally and theoretically. To understand the dynamics of SnCs, we scanned a wide class of mathematical models of SnC dynamics, and compared these models to longitudinal SnC trajectories[1] and direct SnC induction experiments in mice (Fig. 1b–d). The models all describe SnC production and removal. They differ from one another in the way that production and removal rates are affected by age and by SnC abundance. The models describe all combinations of four possible mechanisms for accumulation of SnCs (Fig 1b): (i) SnC production rate increases with age due to accumulation of mutations[27], telomere damage, and other factors that trigger cellular senescence[11], (ii) SnCs catalyze their own production by paracrine and bystander effects[28], (iii) SnC removal decreases with age due to age-related decline in immune surveillance functions[29], and (iv) SnCs reduce their own removal rate, which can be due to SnC-related signaling, such as SASP, down-regulation of immune surveillance by SnCs, SnCs saturating immune surveillance mechanisms (similar to saturation of an enzyme by its substrate), or to disruption of tissue and extracellular matrix architecture that interferes with removal.

Mechanism (iv) is distinct from mechanism (iii) because the decline in removal rate in (iv) depends on SnC abundance, rather than on age directly. Although (iv) can arise from various biological processes, we denote it for simplicity 'saturation of removal'. These four effects lead to 16 different circuits (Fig. 1b) with all combinations of whether or not each of effects (i–iv) occur. Additionally, each of the 16 models includes parameters for basal production and removal. The models have rate constants that are currently uncharacterized. We also tested models which incorporate additional non-linearities (Supplementary Note 1, Supplementary Fig. 1).

## Results

**SnC dynamics during ageing in mice.** To find which of the model mechanisms best describes SnC dynamics, and with which rate constants, we compared the models to longitudinal data on SnC abundance in mice collected by Burd et al. [1]. SnC abundance was measured using a luciferase reporter for the expression of p16[INK4a], a biomarker for SnCs. Total body luminescence (TBL) was monitored every 8 weeks for 33 mice, from early age (8 weeks) to middle–late adulthood (80 weeks) (Fig. 2a).

The luciferase in these mice was introduced into one of the p16 loci, causing the mice to be heterozygous for p16, which may impair proper activation of the senescence program. We therefore

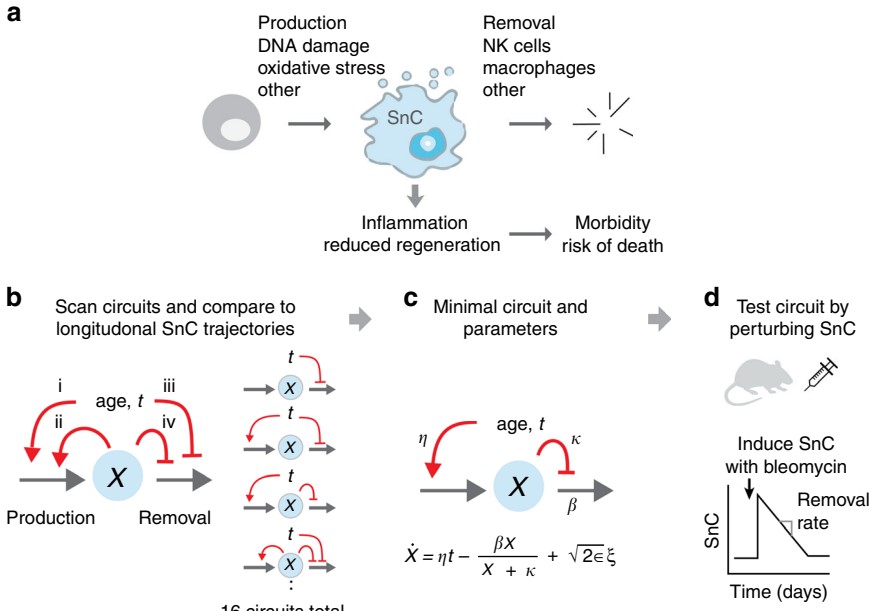

**Fig. 1** Approach for inferring SnC dynamics throughout adulthood. **a** Many processes, including DNA damage and developmental and paracrine signals, lead to SnC production. SnCs are cleared by immune mechanisms, and secrete factors that lead to morbidity and mortality. **b–d** We scanned a wide class of models for SnC dynamics, and compared them to longitudinal SnC data and direct SnC perturbation experiments to establish a minimal model for SnC stochastic dynamics and determine its rate constants. In the minimal model, $\eta$ is the increase in SnC production rate with age, $\beta$ is the removal rate, $\kappa$ is the half-way saturation point for removal, and $\epsilon$ is the noise amplitude.

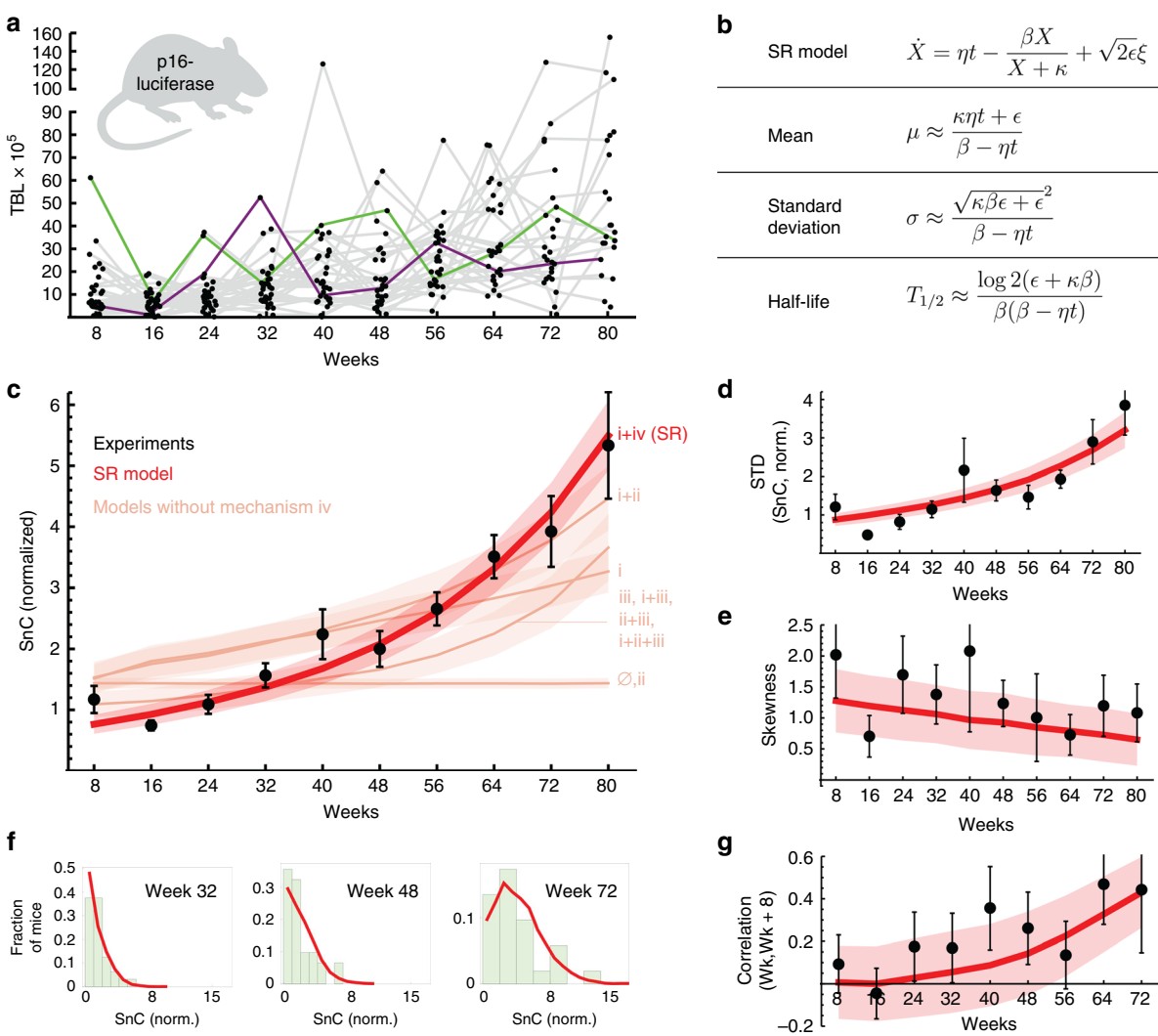

**Fig. 2** Saturated-removal (SR) model captures longitudinal SnC trajectories in mice. **a** Total body luminescence (TBL) of p16-luciferase in mice ($n = 33$). Gray lines connect data from the same individual mice (green and purple lines are examples of individual trajectories). **b** SR model equations and their approximate analytical solutions. The SR model (red line) captures **c** the mean SnC abundance, **d** standard deviation of SnC abundance, **e** skewness, and **f** shape of the distributions among equal-aged individuals, and **g** correlation between subsequent measurements on the same individuals. TBL was normalized to give a mean abundance of 1 at young ages. Maximum-likelihood parameters for the SR model are: $\eta = 0.15\ \text{day}^{-1}\ \text{year}^{-1}$, $\beta = 0.27\ \text{day}^{-1}$, $\kappa = 1.1$, $\epsilon = 0.14\ \text{day}^{-1}$. Pink lines in **c**: best-fit of all models without saturation mechanism iv, that have an age-related increase in SnCs, best-fit parameters are in Supplementary Note 1. Mean and standard error (shaded red, pink regions) are from bootstrapping. Source data are provided as a Source Data file.

also tested longitudinal measurements of SnCs based on another method. For this we obtained longitudinal data from Yamakoshi et al.[30], who measured SnC abundance by creating a transgenic mouse model with a human p16 gene tagged with luciferase, retaining the native p16 loci. Although this dataset has much fewer mice, it shows similar dynamics to the dataset of Burd et al.[1] (Supplementary Note 1, Supplementary Fig. 2), suggesting a similar underlying dynamical process.

We tested how well each model describes the longitudinal SnC trajectories of Burd et al.[1] by finding the maximum-likelihood parameters for each of the 16 models, adjusting for number of parameters (Supplementary Notes 1 and 2, Supplementary Tables 1–4). A principle emerges from this comparison: in order to capture the longitudinal dynamics, the mechanism must have rapid turnover of SnCs on the timescale of a few days in young mice, and it also must include mechanism (iv), which represents a decline in removal that depends on SnC abundance rather than directly on age. The simplest model that describes the data thus has only two

interactions (Fig. 1c): SnC production rate increases linearly with age (mechanism i), and SnCs slow down their own removal rate (mechanism iv). We call this model the saturating removal model (SR model), whose equation is given in Fig. 2b.

The SR model captures the accelerating rise of mean SnC abundance with age in the longitudinal data (Fig. 2c and Supplementary Figs. 3, 4): as SnCs accumulate, they slow their own removal, leading to even higher SnC levels. The SR model also explains the SnC variability between individuals which accelerates with age (Fig. 2d), and the SnC distributions among equal-aged individuals (Fig. 2e), which are skewed to the right (Fig. 2f).

Importantly, the SR model captures the fact that SnC fluctuations become more persistent with age, as evidenced by an increasing correlation between subsequent measurements (Fig. 2g, F-test for linear regression, p-value 0.0047; F-statistic 16.5): individuals with higher (or lower) than average SnC levels stay higher (or lower) for longer periods with age. This increased persistence is due to the effect of SnCs on their own removal rate.

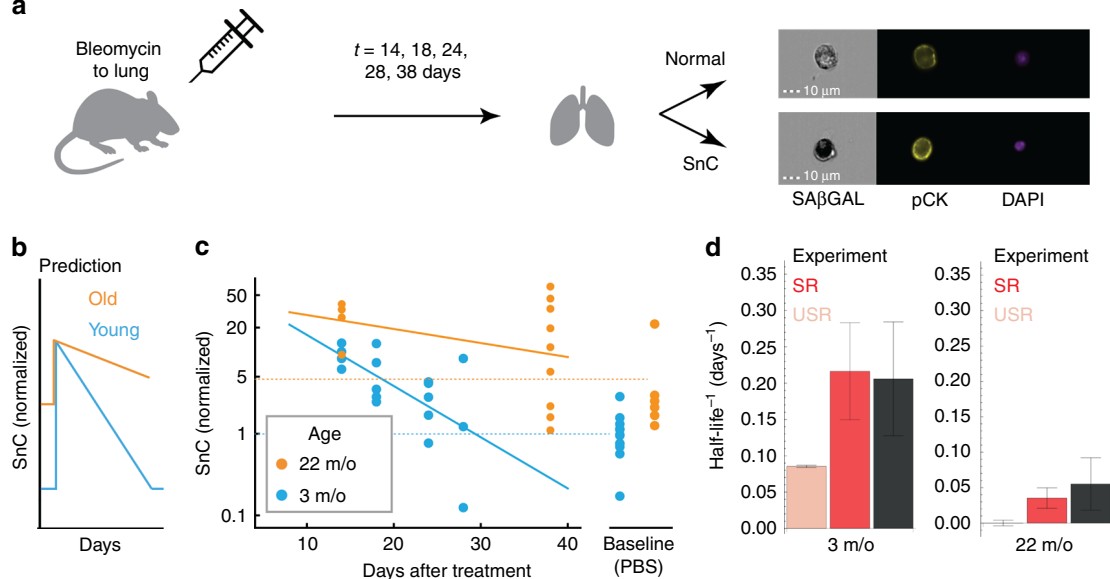

**Fig. 3** SnC half-life measurements in mice support SR model predictions. **a** Bleomycin or PBS was introduced by intratracheal installation to mice on day 0. Lungs were analyzed on the indicated days thereafter. Representative images of lung cells analyzed by imaging flow cytometry show how senescent epithelial cells were identified, using SA-β-Gal, Pan-Cytokeratin (pCK), and DAPI staining. SnC removal rate was estimated by log-linear fit. **b** The SR model predicts that SnCs rapidly return to baseline in young mice and that removal is slower in old mice. **c** Fraction of SnCs in mouse lungs after treatment with bleomycin (1.5 U/kg). In young mice, SnC levels return to baseline with a half-life of about 5 days. In old mice, baseline SnC levels are about five-fold higher, and SnC removal rate is slower than in young mice . **d** SnC removal rates (half-life$^{-1}$) for young and old mice (mean and standard error from bootstrapping, black) agree with the SR model predictions (red, mean and SE were calculated by bootstrapping, see the "Methods" section). The best-fit model without mechanism (iv), the USR model (mechanisms i + iii), shows a poor prediction (pink). For both ages, the USR prediction is different from the observed half-life with $p < 0.01$ from bootstrapping. Source data are provided as a Source Data file.

Models without mechanism iv (saturation of removal) show a poor overall fit (pink lines in Fig. 2c, ΔBIC > 44.3).

**SnC lifetime is days in young mice and weeks in old mice**. The maximum-likelihood parameters of the SR model (listed in the caption of Fig. 2) provide quantitative predictions for SnC half-lives: SnC turnover is rapid in young mice, with a half-life of about 5 ± 1 days at 3 months of age; Turnover slows with age, so that SnC half-life is about 25 ± 6 days at 22 months.

We tested these predictions using experiments in mice by inducing SnCs and analyzing their dynamics. To induce senescence in mice lungs we used intra-tracheal bleomycin administration (Fig. 3a), a DNA-damaging agent that induces cellular senescence in the lung epithelium a few days after treatment[5,31].

We quantified the fraction of senescent lung epithelial cells at different time points following bleomycin administration (Fig. 3a) using imaging flow cytometry. Epithelial SnCs were defined as cells positive for a senescent cell marker (SA-β-Gal) and an epithelial marker (pan-Cytokeratin, pCK). This cell population was also HMGB1 nuclear negative, as expected in SnCs[5,32], and previously shown[5] to correspond to non-proliferative cells (negative Ki67 assay, see Supplementary Note 3, Supplementary Fig. 6).

In 3-month-old mice, SnC levels decayed with a half-life of $\tau = 4.7$ days ($\tau^{-1} = 0.21 +/- 0.07$ days$^{-1}$) and reached their baseline level within less than a month (Fig. 3b, c), as predicted. SnC levels in young mice lungs are thus in a rapid dynamic balance of production and removal.

To test the prediction that removal slows with age (Fig. 3b), we performed the bleomycin treatment in old mice (22-month old). In these mice, the baseline level of SnCs was about five-fold higher than in young mice (Fig. 3d). SnCs decayed with a half-life of

$\tau = 18$ days, $\tau^{-1} = 0.055 +/- 0.035$ days$^{-1}$), slower than that of young mice as predicted ($p = 0.038$ from bootstrapping, Fig. 3b).

These turnover measurements quantitatively agreed with the predictions of the SR model (Fig. 3d, Supplementary Note 4, Supplementary Fig. 7) with no additional fit. This agreement occurred despite the use of distinct SnC markers in the two data sets (SA-β-Gal in the bleomycin experiment vs. p16$^{INK4A}$-luciferase in the longitudinal experiment), suggesting consistency between the measurement methods.

Our results suggest a core mechanism in which SnC production rate rises linearly with age, and SnCs slow their own removal (Supplementary Note 5, Supplementary Fig. 8). This slowdown of removal accelerates SnC accumulation with age. Slowdown of removal also amplifies fluctuations in SnC levels at old ages. This amplification, known as critical slowing down[33,34], results in long-lasting differences among individuals at old ages. In other words, young mice have large spare removal capacity of SnC; old mice have much smaller spare removal capacity. This smaller removal capacity means that any addition of SnCs takes longer to remove, causing larger and more persistent variation in SnC levels among individuals (Fig. 2g).

**The SR model quantitatively recapitulates the Gompertz law**. In the remainder of the paper, we use mathematical analysis to explore the implications of these findings for the question of variability in mortality. Mortality times vary even in inbred organisms raised in the same conditions, demonstrating a non-genetic component to mortality[35,36]. In many species, including mice and humans, risk of death rises exponentially with age, a relation known as the Gompertz law[37–39], and decelerates at very old ages. The Gompertz law has no known explanation at the cellular level.

To connect SnC dynamics and mortality, we need to know the relationship between SnC abundance and risk of death[1]. The precise relationship is currently unknown. Clearly, SnC abundance is not the only cause for morbidity and mortality. It seems to be an important causal factor because removing SnCs from mice increases mean lifespan[25], and adding SnCs to mice increases risk of death and causes age-related phenotypes[23]. We therefore explored the simple possibility that death can be modeled to occur when SnC abundance exceeds a threshold level $X_C$, representing a collapse of an organ system or a tipping point such as sepsis (Fig. 4a). Thus, death is modeled as a first-passage time process, when SnC cross $X_C$. We use this assumption to illustrate our approach, because it provides analytically solvable results. We also show that other dependencies between risk of death and SnC abundance, such as sigmoidal functions with various degrees of steepness, provide similar conclusions.

The SR model analytically reproduces the Gompertz law, including the observed deceleration of mortality rates at old ages (Fig. 4b–d, Supplementary Note 2, Supplementary Fig. 5, Supplementary Table 5). Notably, most models without both rapid turnover and slowdown of removal do not provide the Gompertz law (Supplementary Note 2). The deceleration of mortality rates at very old ages occurs in the model due to the increased persistence of SnC at old age. Those with high SnC have already died, whereas those with low SnC retain low SnC levels for long periods of time and avoid death. The SR model gives a good fit to the observed mouse mortality curve (Fig. 4b, c, Supplementary Note 1) using parameters that agree with the present experimental half-life measurements and longitudinal SnC data (Supplementary Note 1). Thus, turnover of days in the young and weeks in the old provides SnC variation such that individuals cross the death threshold at different times, providing the observed mortality curves.

The SR model can describe the observed increase in mean lifespan of mice in experiments in which a fraction of SnCs are continually removed (Supplementary Note 6). More generally, the SR model can address the use of drugs that eliminate SnCs, known as senolytics[40]. To reduce toxicity concerns, it is important to establish regimes of low dose and large inter-dose spacing[41]. The model provides a rational basis for scheduling senolytic drug administrations. Specifically, treatment should start at old age, and can be as infrequent as the SnC turnover time (~month in old mice) and still be effective (Supplementary Note 6).

We also adapted our results from the mouse data to study human mortality curves. In humans, mortality has a large non-heritable component[42,43]. A good description of human mortality data, corrected for extrinsic mortality, is provided by the same parameters as in mice, except for a 60-fold slower increase in SnC production rate with age in the human parameter set (Fig. 4d, Supplementary Note 7, Supplementary Table 6). This slower increase in SnC production rate can be due to improved DNA maintenance in humans compared to mice[44]. We conclude that the critical slowing-down described by the SR model provides a possible cellular mechanism for the variation in mortality between individuals.

**SR-type dynamics and ageing of *Drosophila* and *C. elegans*.** The generality of the SR model suggests that it might also apply to organisms where ageing may be driven by factors other than SnCs, such as *Drosophila melanogaster* and *C. elegans*, in which lifespan variation is well-studied[35,45]. In these organisms, the present approach can be extended by considering $X$ as a causal factor for aging, that accumulates with age and has SR-type dynamics[46], namely turnover that is much more rapid than the lifetime, increasing production and self-slowing removal. One clue for the identity of such factors may be gene-expression variations in young organisms that correlate with individual lifespan[47–49], and the actions of genes that modulate lifespan[39,50–53].

Work in *C. elegans* and *Drosophila* provides constraints to test the SR model. For example, *Drosophila* shows rapid switches between hazard curves when transitioned between normal and lifespan-extending diets (Fig. 4e, inset). These switches are well-described by the SR model, due to its rapid turnover property (Fig. 4e and Supplementary Note 8, Supplementary Fig. 9). The rapid turnover property entails that the level of $X$ can adjust after a change in any of the parameters of the model. A model without rapid turnover could not explain these results.

We further tested whether the SR model can explain the survival curves of *C. elegans* under different life-extending genetic, environmental, and dietary perturbations[35]. These perturbations change mean lifespan by up to an order of magnitude. The survival curves show a remarkable feature called temporal scaling: the survival curves collapse onto approximately the same curve when age is scaled by mean lifespan (Fig. 4f insets). That is, the entire distribution of death times, including its mean and standard deviation, is determined by a single parameter, which depends on the perturbation. We find that the SR model provides the shape of the survival curves, as well as their temporal scaling feature. Temporal scaling is found in the SR model by assuming that the perturbations affect the accumulation rate $\eta$ (Fig. 4f, Supplementary Note 9 and Supplementary Fig. 10A).

Temporal scaling cannot be explained by models without rapid turnover (Supplementary Fig. 10B), or by varying any other parameter except $\eta$ in the SR model. Thus, we predict loss of temporal-scaling of survival curves when a perturbation affects other SR-model parameters, such as removal rate $\beta$ or noise $\in$ (Supplementary Fig. 10C, D). This prediction may apply to exceptional perturbations in which temporal scaling is not found, such as the *eat-2* and *nuo-6* mutations (Supplementary Fig. 10E, F). We conclude that the SR model of rapid turnover with critical-slowing down is a candidate explanation for the temporal scaling of survival curves in *C. elegans*.

## Discussion

In this study, we propose a framework for the dynamics of SnCs based on rapid turnover that slows with age. Bleomycin-induced SnC half-life is days in young mice and weeks in old mice, causing critical slowing down, which greatly amplifies the differences between individual SnC levels at old age. We theoretically explore the implications of this slowdown in a model in which SnCs cause death when they exceed a threshold. The widening variation in SnC levels with age causes a mortality distribution that follows the Gompertz law of exponentially increasing risk of death. The mortality distribution of mice and humans is well-described by the SR model with the SnC half-lives measured here. Future work may test this proposed connection between SnC dynamics and mortality by experimentally measuring risk of death as a function of SnC abundance.

The rapid removal of SnCs that we observe following bleomycin-induced DNA damage is in line with studies that showed efficient removal of SnCs in vivo following liver fibrosis or induction by senescence by mutant Ras[54–56]. On the other hand, when senescence was induced in the skin by directly activating the cell-cycle inhibitor *p14ARF*, which was not associated with an increase in tissue cytokine expression or inflammation, the induced SnCs persisted in the tissue for several weeks[57].

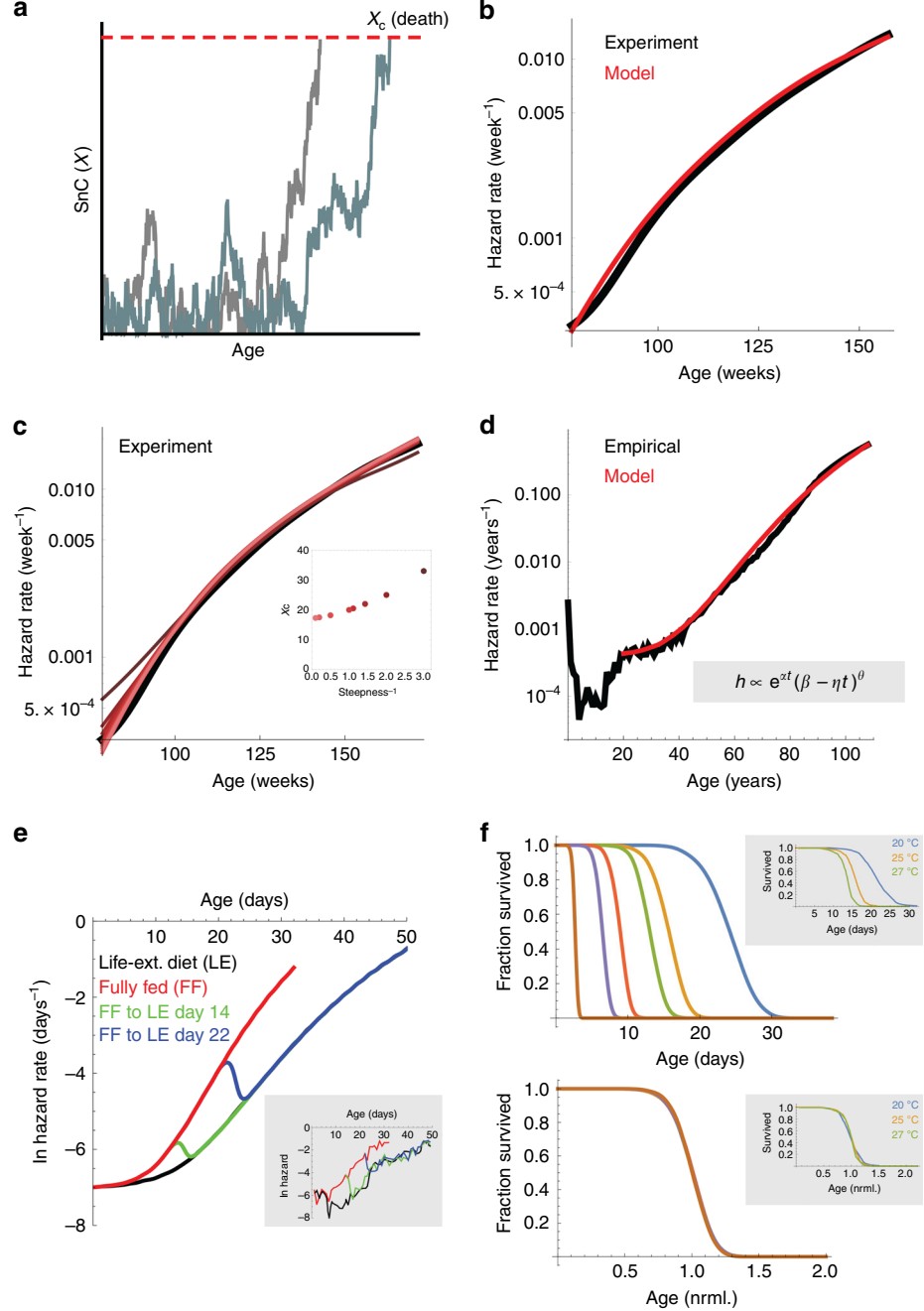

**Fig. 4** SR model can explain the variability in mortality between individuals. **a** To model the relation between risk of death and SnC levels, we assumed a simple threshold model where death occurs when SnC abundance exceeds a critical threshold $X_C$. **b** Mouse mortality (C57BL/6J mice obtained from the Mouse Phenome Database[60], black line) is well fit by the SR model (red line) with parameters consistent with the data of Figs. 1, 2, with death defined when SnC exceed a threshold ($\eta = 0.084$ day$^{-1}$ year$^{-1}$, $\beta = 0.15$ day$^{-1}$, $\kappa = 0.5$, $\varepsilon = 0.16$ day$^{-1}$, $X_C = 17$). **c** Similar results are obtained by assuming a more general sigmoidal dependency between SnC abundance $X$ and risk of death: $h = \left(1 + e^{-\theta(X-X_C)}\right)^{-1}$. Parameters are the same as **b**, except that $X_C$ is adjusted according to the steepness parameter θ (inset). **d** The SR model with added age-independent extrinsic mortality of $0.4 \times 10^{-3}$ year$^{-1}$ (red) matches human mortality statistics[61] (black). Inset: approximate analytical solution for the first passage time in the SR model shows the Gompertz law and deceleration at old ages. The parameters are similar to **b**, except a ~60-fold decrease in $\eta$: $\eta = 0.00135$ day$^{-1}$ year$^{-1}$, $\beta = 0.15$ day$^{-1}$, $\kappa = 0.5$, $\varepsilon = 0.142$ day$^{-1}$, $X_C = 17$. **e** SR model describes rapid shifts in mortality when fully fed *Drosophila* transition to a lifespan-extending dietary intervention (LE), (inset: experimental data from Mair et al.[45]), with $\beta = 1$ h$^{-1}$, $\kappa = 1$, $\varepsilon = 1$ h$^{-1}$, $\eta = 0.03$ day$^{-1}$ h$^{-1}$ and $X_C = 15$. LE was modeled by a decrease in $\eta$: $\eta = 0.02$ h$^{-1}$ day$^{-1}$ (changes in other parameters lead to similar conclusions, see Supplementary Note 8). **f** Lifespan of *C. elegans* raised at different temperatures varies by an order of magnitude, but survival curves collapse on a single curve when time is scaled by mean lifespan (inset: data from Stroustrup et al. [35]). The SR model provides scaling for perturbations that affect $\eta$, but not other parameters ($\beta = 1$ h$^{-1}$, $\kappa = 1$, and $\varepsilon = 1$ h$^{-1}$, $\eta = 0.07$ h$^{-1}$ day$^{-1}$, $X_C = 20$, Supplementary Note 9). Source data are provided as a Source Data file.

Clearance may thus depend on the tissue, on the method of senescence induction, and on the presence of SASP.

The present analysis of longitudinal p16 trajectories suggests that SnC slow down their own removal rate. This effect may be due to several mechanisms, including SASP, disruption of tissue architecture, or SnC abundance exceeding immune capacity. For the latter effect, SnC abundance at old age needs to be comparable to the abundance of the immune cells that remove them, which make up on the order of 0.1% of the body's cells[58,59]. Further research is needed to characterize these effects.

Our results suggest that treatments that remove SnCs can therefore have a double benefit: an immediate benefit from a reduced SnC load, and a longer-term benefit from increased SnC removal. Similarly, interventions that increase removal capacity, for example by augmenting the immune surveillance of SnC, are predicted to be an effective approach to reduce SnC levels. More generally, the present combination of experiment and theory can be extended to explore further stochastic processes in aging, in order to bridge between the population-level and molecular-level understanding of aging.

## Methods

**Stochastic model simulation**. Simulations of the stochastic models were performed by using the ItoProcess function of Mathematica (V11.3), with a step size of 1 day. Negative $X$ values were avoided using a reflecting boundary condition at $X = 0$. In simulations that included mortality, time of death was the first time-point where $X$ exceeded $X_C$.

**Model comparison to p16INK-luciferase measurements**. We sought for each model the parameters that maximize the log-likelihood of the measured trajectories (Fig. 1a). We calculated the log-likelihood of a model $m$ with parameters $\theta$ as follows. Let $X_{i,j}$ be the measured SnC level (SnC = TBL/9.63 to give SnC = 1 for young mice) of mouse $j$ at time point $i$ (with $X_{0,j} = 0$). We denote by $\text{Prob}_{m,\theta,i}(a|b)$ the probability of reaching SnC level $a$ at time point $i$ given SnC level $b$ at time point $i-1$. We call such a step from $i-1$ to $i$ a sub-trajectory. We estimated this probability from simulations (4000 simulations for every such sub-trajectory). The log-likelihood is $\text{LL}(m,\theta) = \sum_j \sum_i \log(\text{Prob}_{m,\theta,i}(X_{i,j}|X_{i-1,j}))$, $n = 294$ sub-trajectories. For each model, we sought the parameter set that maximizes the log-likelihood (see Supplementary Note 1 for more details). Confidence intervals for the best-fit parameters, as well as for estimates for SnC half-life, were calculated by bootstrapping (selecting mice at random with replacements). Modeling experimental noise (multiplicative noise with amplitude up to 30%) did not affect the best-fit parameters (Supplementary Note 1). To find parameters for the model that describe both the longitudinal trajectories as well as mouse mortality statistics, we scanned the subset of parameters that fit the mortality distribution of mice. Mortality statistics of WT (C57BL/6J) mice were obtained from the Mouse Phenome Database[60]. Because mortality in young mice appears to be unassociated with the accumulation of SnCs[1], we only considered deaths that occurred after age one year (which make up 97% of the total deaths in the dataset). We performed a comprehensive scan of values of $\beta_0$ and $\kappa_0$ and constrained $\eta, \in$ to values that give a mean and standard deviation of the simulated mortality distributions within 2% of the empirical values for WT mice. The critical SnC level $X_C$ was set at $X_C = 17$, which is the maximal SnC level in the Burd et al. dataset[1]. We then calculated maximum-likelihood parameters and confidence intervals as described above.

**Population-level measures**. The mean and variance at time-point $i$ are the mean and variance of $\{X_{i,j}\}_{j=1}^N$, where $N$ is the number of mice. Autocorrelation is the Pearson correlation of the two vectors $\{X_{i,j}\}_{j=1}^N$, $\{X_{i+1,j}\}_{j=1}^N$. The measures were calculated in the same manner from model simulations. Typical SnC removal rate (half-life$^{-1}$) for the model at a given age $i$ was estimated by $\frac{\beta}{\kappa + \overline{X}_i} \log(2)^{-1}$, where $\overline{X}_i$ is the mean SnC level at age $i$ (see Supplementary Note 4 for discussion of alternative ways to estimate SnC half-life). For the USR model, typical SnC removal rate at age $i$ is $(\beta_0 - \beta_1 i) \log(2)^{-1}$.

**Quantification of SnCs in mouse lung epithelium**. We subjected 3-month-old (young) and 22-month-old (aged) C57BL6 mice to intra-tracheal installation of 1.5 U/kg bleomycin (Sigma) solution in PBS (or PBS as a control treatment). We euthanized the mice at 14, 18, 24, and 28 days for the young mice and day 14 and day 38 for the aged mice. The quantification of senescent epithelial cells was performed as previously described[5] with modification. Lung tissue was chopped into 2–5 mm pieces in HBSS (14025050, Gibco) on ice and incubated in the 5 ml dissociation buffer (1 mg/ml Collagenase Type IV (C9263, Sigma), 0.1 mg/ml DNase I (10104159001, Roche) in HBSS) at 37 °C for 50 min. Cells were washed

with HBBS and then fixed with 4% PFA for 5 min. Post fixation, cells were washed and incubated with X-Gal-staining solution for 16 h at 37 °C. The X-Gal-staining solution consisted of 5 mM K$_3$Fe(CN)$_6$, 5 mM K$_4$Fe(CN)$_6$ × 3H$_2$O and 2.5 mM X-Gal (Inalco) in PBS at pH 5.5 containing 1 mM MgCl$_2$. Post X-Gal staining the cells were fixed with fixation buffer for 30 min at 4 °C and washed with permeabilization buffer (00-5223-56, eBioscience, San Diego, CA). The cells were then incubated with PE-conjugated pan-cytokeratin (ab52460, Abcam) and HMGB1 (ab18256, Abcam) antibodies for an hour at 4 °C. For visualization of HMGB1 antibody, we used Qdot605-labeled Goat Anti-Rabbit antibody (Q11402MP, ThermoFisher). Antibodies were diluted in the permeabilization buffer with the dilution of 1:100 of PE-conjugated pan-cytokeratin and primary HMGB1, and 1:50 of Qdot605. Before visualization, the cells were stained with DAPI and filtered through a 100 μm membrane. The resulting cells were analyzed by imaging flow-cytometry using ImageStreamX mark II (Amnis, Part of EMD milipore—Merck, Seattle, WA, USA, see Supplementary Note 3 for gating strategy summary). PE staining was collected at channel 3, the DAPI at channel 7 and the Qdot605 at channel 10, in addition to the bright-field images collected at channels 1 and 9. Analysis of the image data was performed using IDEAS 6.2 software. Cells were first gated according to their area (in μm$^2$) and aspect ratio (ratio between width and length) of the bright field images, to eliminate debris and aggregates. Then, we gated on focused cells using the gradient RMS (which measures the sharpness quality of an image by using the average gradient of a pixel normalized for variations in intensity levels) and contrast (measures the sharpness quality of an image by detecting large changes of pixel values). Cropped cells were excluded by using the centroid X feature (the number of pixels in the horizontal axis from the upper, left corner of the image to the center of the image mask). To verify that only single cells were analyzed, cells were further gated for single nuclei using the area and aspect ratio of the nuclear image of the DAPI staining. SA-beta-Gal staining was quantified using the Mean pixel (the mean of the background-subtracted pixels) contained in the bright-field image[5]. Staining of pCK was quantified using the Intensity (the sum of the background subtracted pixel values within the image) and the Max pixel (the largest value of the background-subtracted pixels contained in the image) features of the corresponding channels. To quantify staining of HMGB1 specifically, its intensity was calculated. We first gated for pCK-positive cells then for HMGB1 negative, SA-β-Gal-positive cells to quantitate the SnCs in lung epithelium. Following the method, establishment of pCK positive, SA-β-Gal-positive cells were considered senescent in further experiments. In total, the experiments were 3-month-olds treated with 1.5 U/kg ($n = 17$), 3-month-olds treated with PBS ($n = 13$), 22-month-olds treated with 1.5 U/kg ($n = 13$), and 22-month-olds treated with PBS ($n = 6$). We complied with all relevant ethical regulations for animal testing and research. The experiments were approved by the Weizmann IACUC committee.

**Analysis of the bleomycin treatment time series**. We estimated the turnover of SnCs by calculating the removal time after a perturbation with bleomycin. The removal rate is the slope of the log-linear regression model, which we fit for each experiment (with confidence intervals calculated by bootstrapping). We obtained the response time predicted by the model by bootstrapping and simulating the model after perturbation (see Supplementary Note 4 for details). Statistical significance tests were computed by bootstrapping.

**Estimation of hazard and survival functions**. We fit hazard and survival functions from mortality data by interpolation using the Mathematica (V11.3) function SmoothKernelDistribution and then applying the Mathematica functions HazardFunction and SurvivalFunction. For the mice survival data, the SmoothKernelDistribution was computed with a bandwidth of 80 days.

**Simulation of Drosophila and C. elegans survival curves**. We simulated the mortality trajectories of Drosophila and C. elegans using the SR model, by assuming a rapid turnover and saturation $\beta = 1\,h^{-1}$, $\kappa = 1$ [au], and also set $\varepsilon = 1$ [au]$^2\,h^{-1}$ where [au] is the mean level of $X$ in young organisms. These parameters correspond to a turnover of $X$ on the order of hours. For C. elegans, to fit the survival curve obtained by Stroustrup et al.[35], we set $\eta = 0.07$ [au] h$^{-1}$ day$^{-1}$ and assumed that death occurs when $X > X_C$ for $X_C = 20$ [au]. Similarly, to fit the Mair et al.[45] data, we set the following parameters for the Drosophila simulations: $\eta = 0.03$ [au] h$^{-1}$ day$^{-1}$ and $X_C = 15$ [au], and assumed a baseline mortality of ln hazard $= -7$ day$^{-1}$.

**Reporting summary**. Further information on research design is available in the Nature Research Reporting Summary linked to this article.

## Data availability

The source data underlying Figs. 2a, c, d–g, 3c, 4b–d, and Supplementary Fig. 2A is available as a Source File. All other data are available from the corresponding author upon request.

## Code availability

Custom code was written in Mathematica 11.3 (Wolfram) and can be found in the following link: https://github.com/omerka-weizmann/sncdynamics. The analysis file

contains functions for simulating circuit topologies, obtaining mortality statistics and survival-curve scaling, computing log-likelihood of stochastic trajectories, and computing various summary statistics of stochastic trajectory data.

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

## Acknowledgements

U.A. is the incumbent of the Abisch-Frenkel chair. O.K. is an Azrieli Fellow. This work was supported by grants to V.K. from the European Research Council under the European Union's FP7 and H2020 Programs and the Israel Science Foundation.

## Author contributions

Conceptualization, O.K., U.A. and V.K.; Planning and setup of mouse experiments, A.A., V.K., Execution of experiments: A.A., Analysis of ImageStreamX data, A.A, Z.P.; Derivation of theory, O.K. and U.A.; Data analysis and modeling, O.K.; Writing manuscript, O.K. and U.A.; Review and editing: O.K., A.A., U.A. and V.K.

## Competing interests

The authors declare no competing interests.
