## [Peer Review File · Nature Communications]

Reviewers' comments:

Reviewer #1 (Remarks to the Author):

This is a very interesting paper, with an original and valuable approach to the understanding of senescence in vivo and its contribution to ageing.

I have some concerns that I think the authors could consider before publication:

1. The experimental data used to generate the mathematical model come from a mouse that is heterozygous for p16 (Burd et al. 2013). If we assume that p16 is a good read out of senescence, then, the mice used are only monitoring half the amount of p16; also if we assume that p16 is important for the implementation of senescence, then, the mice used are partially impaired in the activation of senescence. I think that the authors should acknowledge these caveats and should address them.
2. There is another mouse model that retains normal endogenous p16, but carries a transgenic allele of human p16 fused to luciferase (the resulting human p16-luc protein is inactive regarding hp16 but active regarding luc) (Yamakoshi et al., 2009: <https://www.ncbi.nlm.nih.gov/pubmed/19667129>). This model also has caveats of its own, of course, but are different caveats from those of Burd et al. I wonder if the authors could obtain the primary data from the longitudinal study in Yamakoshi and see how well it adjust to their model. It would be a nice validation using a set of data external to the set of data used to train the model.
3. I don't fully grasp the difference between models (iii) and (iv). In (iii), removal slows down because the immune system slows down, but this reduced activity of the immune system can be consequence of the paracrine actions of senescent cells, which then sounds very similar to model (iv).
4. Is model (iv) the same as the "saturation removal" (SR) model? Please, clarify the difference between (iii), (iv) and SR. In lines 123 and 141, authors refer to their model as: "rapid SnC turnover and removal slowdown". I like this way of referring to their model because I understand it immediately. However, I am not sure if this is (iii), (iv) or SR, or none? I suggest that if authors have no data on "saturation" of the immune system, that they simply say "impaired immuneclearance" (by saturation, or because senescent cells make themselves invisible, or because the extracellular matrix is impenetrable, or because the bone marrow does not produce healthy active immune cells, etc).
5. I find unlikely that senescent cells may reach such high levels in old mice as to saturate the immune system (innate or adaptive). Is there any evidence for this?
6. In Fig. 2C, which are the models without saturation removal?
7. How good or bad is model (iii) compared to (iv)?
8. In Fig. 3D: authors show how bad it is the best fit model without SR. Again, model (iii), how bad is it?
9. I miss discussion of other papers that have looked at the rate of senescent cell elimination in the epidermis and have found that senescent cells are persistent (<https://www.ncbi.nlm.nih.gov/pubmed/23423975>).
10. My particular interpretation of the Gompertz effect is that those individuals with high fluctuations (in this case senescent cells) are those that die first, so that there is a selection for those that fluctuate less and this reflects as a slow down in death risk. Would the authors agree with this? I am trying to suggest an interpretation that is easier to communicate.

Reviewer #2 (Remarks to the Author):

Karin et al. present analysis of the dynamics of senescent cells, by constructing a suite of mathematical models to describe senescence and performing model selection, using data on senescence in mice, which identifies certain model characteristics that are key for describing senescence. This reveals a simple and compelling mechanism by which senescent cells accumulate (the removal rate of senescent cells, which is self-regulated, decreases with age) that fits the data well, and independent experiments are found to be in quantitative agreement. This is an interesting study that will be of interest to a broad community. Below I list a number of questions

or requests for clarifications.

1. The “saturating removal” effect introduces the only nonlinearity in X that is considered in this model: does this go some way to explain the fit. One could imagine other nonlinearities, for example a logistic growth term (rather than a linear term) in η_2 . Would such a model fit the data? The best-fit model approximates a Gompertzian curve for age-related mortality: is this also in part simply due to the form of the saturating removal term. Another way to put this question would be: how confident are the authors that they have explored the space of possible models?

2. In Fig. 1 it would be very helpful to label the arrows of the model schematic with their associated parameters.

3. In Fig. 2, at 80 weeks the variance in TBL is large, with many mice still exhibiting low levels. Is it by chance that the two trajectories shown in Fig. 2A do not lie close to the mean trajectory of the data? It would be helpful to plot more (all?) of these trajectories overlapping, to get a better sense of these. It seems that many mice do not show increases in TBL over their lifetimes. This needs further investigation, as it may suggest that while the SR model fits the mean SnC dynamics, trajectories for individual mice may be fit better by alternative mechanisms?

4. Lines 181-186: hard to follow. Please expand on this section as it is unclear. Also, here (and elsewhere) it can be difficult to follow the references to the SI - please make these more specific wherever possible, e.g. to a specific SI figure rather than to a section.

5. Supplementary Section 2 Eq. 3 needs further justification/clarification: should the term $\beta \kappa \log(\kappa + X)$ be $\beta X \log(\kappa + X)$? This has implications for the following quasi-steady state analysis.

6. Please clarify the meaning of the term “ExpIntegralE” in the SI.

Reviewers' comments:

Reviewer #1 (Remarks to the Author):

This is a very interesting paper, with an original and valuable approach to the understanding of senescence in vivo and its contribution to ageing.

We thank the reviewer for this endorsement.

I have some concerns that I think the authors could consider before publication:

1. The experimental data used to generate the mathematical model come from a mouse that is heterozygous for p16 (Burd et al. 2013). If we assume that p16 is a good read out of senescence, then, the mice used are only monitoring half the amount of p16; also if we assume that p16 is important for the implementation of senescence, then, the mice used are partially impaired in the activation of senescence. I think that the authors should acknowledge these caveats and should address them.

2. There is another mouse model that retains normal endogenous p16, but carries a transgenic allele of human p16 fused to luciferase (the resulting human p16-luc protein is inactive regarding hp16 but active regarding luc) (Yamakoshi et al., 2009: <https://www.ncbi.nlm.nih.gov/pubmed/19667129>). This model also has caveats of its own, of course, but are different caveats from those of Burd et al. I wonder if the authors could obtain the primary data from the longitudinal study in Yamakoshi and see how well it adjust to their model. It would be a nice validation using a set of data external to the set of data used to train the model.

We thank the reviewer for these comments and suggestions. We addressed both comments by providing new analysis and additional data in the revised results and SI. We obtained primary data from Yamakoshi et al, as suggested, and find that it shows similar SnC dynamics to the Burd et al dataset. We now address these points in the main text with a new paragraph:

“The luciferase in these mice was introduced into one of the p16 loci, causing the mice to be heterozygous for p16, which may impair proper activation of the senescence program. We therefore also tested longitudinal measurements of senescent cells based on another method. For this we obtained longitudinal data from Yamakoshi et al.³¹, which measured SnC abundance by creating a transgenic mouse model with a human p16 gene tagged with luciferase, retaining the native p16 loci. Although this

dataset has much fewer mice, it shows similar dynamics to the dataset of Burd et al. (Supplementary Section 1), suggesting a similar underlying dynamical process.”

Analysis of the data from Yamakoshi et al. is presented as a new section in the SI.

Supplementary Section 10. Longitudinal trajectories from Yamakoshi et al of mice carrying a transgenic allele of human p16 fused to luciferase with native p16 loci.

Figure S2. Longitudinal mouse luciferase trajectories from a human p16 construct in the presence of native p16 loci. (A) Total body luminescence (TBL) in transgenic mice with the human p16 construct tagged with luciferase from Yamakoshi et al. (12). Each curve represents an individual mouse. (B) For the female mice in the Yamakoshi et al. dataset, both the mean and the standard deviation increase at a similar rate to that of the Burd et al. dataset. (C) Mean TBL rank of female mice in the Yamakoshi datasets, before/after the age 10 months. The trajectories of the female mice in the Yamakoshi dataset show higher persistence at older ages (individuals with high TBL stay high).

Quantitative modelling of the dynamics of SnCs during ageing requires longitudinal measurements of SnCs in individual mice *in vivo*. In the main text, we analyzed such a dataset collected by Burd et al. (2), which included luciferase-based measurements of SnC abundance in 33 individual mice, from early age (8 weeks) to middle-late adulthood (80 weeks). The measurements were based on a knock-in allele, in which the firefly luciferase gene was targeted into one of the endogenous p16 loci. While the luciferase output retains the cis-regulatory elements of p16, the resulting mouse is heterozygous for p16. We therefore tested longitudinal measurements of p16 based on another method.

For this, we obtained data from Yamakoshi et al. (12), which created a transgenic mouse model with a human p16 gene tagged with luciferase, maintaining the native p16 loci. The dataset contains 7 mice (3 male and 4 female), whose total body luminescence was monitored every month (Figure S2A). Luciferase output increased with age for these mice. Unlike in the Burd et al. dataset, there were marked differences in p16^{LUC} between male and female mice in this dataset. While young females in this dataset had very low luciferase output (as in the Burd dataset), young males had luciferase output that was already similar to that of old females. The dataset for the males also had many missing values, with a third of the time-points containing only a single mouse. For this reason, we chose not to aggregate the measurements and to focus our analysis on the female mice.

The Yamakoshi dataset is much smaller than the dataset of Burd et al. (4 mice compared with 33 mice), and we therefore cannot fit it with a dynamical model. However, it resembles the Burd et al. dataset in several important quantitative aspects. The mean luciferase measurements increase at a similar rate with age to the Burd et al. dataset ($10\% \pm 2\%$ per month compared with $10\% \pm 1\%$), as does the standard deviation ($12\% \pm 3\%$ per month compared with $9\% \pm 1\%$) (Figure S2B). In addition, as predicted by the SR model, the Yamakoshi et al. mice show more persistent variation at old ages than at young ages. This can be quantified by the mean rank, which mixes rapidly in the young (≤ 10 mo) showing a mean rank around 2-3, but stays more persistent in the old (> 10 mo): the individuals with high SnC remain high and those with low SnC remain low, Figure S2C. These similarities indicate that the Burd et al and Yamkoshi et al constructs may report similar underlying dynamics.

3. I don't fully grasp the difference between models (iii) and (iv). In (iii), removal slows down because the immune system slows down, but this reduced activity of the immune system can be consequence of the paracrine actions of senescent cells, which then sounds very similar to model (iv).

4. Is model (iv) the same as the "saturation removal" (SR) model? Please, clarify the difference between (iii), (iv) and SR. In lines 123 and 141, authors refer to their model as: "rapid SnC turnover and removal slowdown". I like this way of referring to their model because I understand it immediately. However, I am not sure if this is (iii), (iv) or SR, or none? I suggest that if authors have no data on "saturation" of the immune system, that they simply say "impaired immuneclearance" (by saturation, or because senescent cells make themselves invisible, or because the extracellular matrix is impenetrable, or because the bone marrow does not produce healthy active immune cells, etc).

These two comments helped us to clarify the difference between (iii) and (iv). We now added the following explanation of this point in the introduction:

“(iii) SnC removal decreases with age due to age-related decline in immune surveillance functions³⁰, and (iv) SnCs reduce their own removal rate, which can be due to SnC-related signaling such as SASP, downregulation of immune surveillance by SnCs, SnCs saturating immune surveillance mechanisms (similar to saturation of an enzyme by its substrate), or to disruption of tissue and extracellular matrix architecture that interferes with removal. “

“Mechanism (iv) is distinct from mechanism (iii) because the decline in removal rate in (iv) depends on SnC abundance, rather than on age directly. Although (iv) can arise from various biological processes, we denote it for simplicity ‘saturation of removal’.”

We further clarified the meaning of the SR model by the following revised text in the results:

“A principle emerges from this comparison: in order to capture the longitudinal dynamics, the mechanism must have rapid turnover of SnCs on the timescale of a few days in young mice, and it also must include mechanism (iv), which represents a decline in removal that depends on SnC abundance rather than directly on age. The simplest model that describes the data thus has only two interactions (Figure 2B): SnC production rate increases linearly with age (mechanism i), and SnCs slow down their own removal rate (mechanism iv). We call this model the saturating removal model (SR model), whose equation is given in Figure 2B.”

5. I find unlikely that senescent cells may reach such high levels in old mice as to saturate the immune system (innate or adaptive). Is there any evidence for this?

We thank the reviewer for this question. As far as we know it is unclear what the absolute abundance of senescent cells is in old tissues. One can estimate how high it needs to be in order to saturate the immune system: the known cell types that remove SnCs include NK cells and macrophages, which together make up on the order of 0.1% of the body’s cells. Thus, if senescent cells turn out to approach these levels at old ages, saturation is possible. Otherwise, saturation is unlikely and slowdown of removal may be due to other mechanisms such as SASP. We now added a paragraph in the discussion which addresses it:

“The present analysis of longitudinal p16 trajectories suggests that SnC slow down their own removal rate. This effect may be due to several mechanisms, including SASP, disruption of tissue

architecture, or SnC abundance exceeding immune capacity. For the latter effect, SnC abundance at old age needs to be comparable to the abundance of the immune cells that remove them, which make up on the order of 0.1% of the body's cells^{59,60}. Further research is needed to characterize these effects.“

6. In Fig. 2C, which are the models without saturation removal?

We now clarify this in the figure legend of Figure 2, and by adding the interactions i-iv present in each of the models:

Figure 2. Saturated-removal (SR) model captures longitudinal SnC trajectories in mice. (A) Total body luminescence (TBL) of p16-luciferase in mice. Grey lines connect data from the same individual mice (green and purple lines are examples of individual trajectories). (B) SR model equations and their approximate analytical solutions. The SR model (red line) captures (C) the mean SnC abundance, (D) standard deviation of SnC abundance, (E) skewness and (F) shape of the distributions among equal-aged individuals, and (G) correlation between subsequent measurements on the same individuals. TBL was normalized to give a mean abundance of 1 at young ages. Maximum likelihood parameters for the SR model are: $\eta = 0.15 \text{ day}^{-1}\text{year}^{-1}$, $\beta = 0.27 \text{ day}^{-1}$, $\kappa = 1.1$, $\epsilon = 0.14 \text{ day}^{-1}$. Orange lines in (C): best-fit of all models without saturation mechanism iv, that have an age-related increase in SnCs, best-fit parameters are in Supplementary Section 1. Mean and standard error (shaded red, orange regions) are from bootstrapping.

7. How good or bad is model (iii) compared to (iv)?

To address this question, we compared the models using Bayesian information criterion (BIC) and added the results of this comparison to the main text and to the supplementary section 1. Very strong evidence is considered to be ΔBIC above 10. The ΔBIC between the SR model and the models with mechanism iii (and without iv) is at least 44.3, suggesting very strong support for mechanism iv compared to iii. We also show in Fig 2C the best-fit curves to the Burd dataset with all models that have (iii) instead of (iv), and detail what interactions the models have next to their curves, for easy comparison.

8. In Fig. 3D: authors show how bad it is the best fit model without SR. Again, model (iii), how bad is it?

In Fig 3D we compare the SR model (mechanism i+iv) to the equivalent model with mechanism iii (mechanism i+iii). The latter mechanism shows best-fit parameters that are not compatible with the observed turnover times ($p < 0.01$). We added the following sentence to the legend of figure 3:

The best-fit model without mechanism (iv), the USR model (mechanisms i+iii), shows a poor prediction (orange). For both ages, the USR prediction is different from the observed half-life with $p < 0.01$.

9. I miss discussion of other papers that have looked at the rate of senescent cell elimination in the epidermis and have found that senescent cells are persistent (<https://www.ncbi.nlm.nih.gov/pubmed/23423975>).

We now add this to the discussion:

The rapid removal of senescent cells that we observe following bleomycin-induced DNA damage is in line with studies that showed efficient removal of senescent cells in-vivo following liver fibrosis or induction by of senescence by mutant Ras⁵⁵⁻⁵⁷. On the other hand, when senescence was induced in the skin by directly activating the cell-cycle inhibitor p14ARF, which was not associated with an increase in tissue cytokine expression or inflammation, the induced senescent cells persisted in the tissue for several weeks⁵⁸. Clearance may thus depend on the tissue, on the method of senescence induction, and on the presence of SASP.

10. My particular interpretation of the Gompertz effect is that those individuals with high fluctuations (in this case senescent cells) are those that die first, so that there is a selection for those that fluctuate less and this reflects as a slow down in death risk. Would the authors agree with this? I am trying to suggest an interpretation that is easier to communicate.

We thank the reviewer for this suggestion. We agree that deceleration of aging occurs because of selection of those with low SnC. The intuitive reason involves the extended persistence of SnC due to the slowdown of removal. We now added this to the results:

The deceleration of mortality rates at very old ages occurs in the model due to the increased persistence of SnC at old age. Those with high SnC have already died, whereas those with low SnC retain low SnC levels for long periods of time and avoid death.

Reviewer #2 (Remarks to the Author):

Karin et al. present analysis of the dynamics of senescent cells, by constructing a suite of mathematical models to describe senescence and performing model selection, using data on senescence in mice, which identifies certain model characteristics that are key for describing senescence. This reveals a simple and compelling mechanism by which senescent cells accumulate (the removal rate of senescent cells, which is self-regulated, decreases with age) that fits the data well, and independent experiments are found to be in quantitative agreement. This is an interesting study that will be of interest to a broad community.

We thank the reviewer for this endorsement.

Below I list a number of questions or requests for clarifications.

1. The “saturating removal” effect introduces the only nonlinearity in X that is considered in this model: does this go some way to explain the fit. One could imagine other nonlinearities, for example a logistic growth term (rather than a linear term) in η_2 . Would such a model fit the data? The best-fit model approximates a Gompertzian curve for age-related mortality: is this also in part simply due to the form of the saturating removal term. Another way to put this question would be: how confident are the authors that they have explored the space of possible models?

To address this, we scanned alternative models with nonlinearities other than saturation, including the suggested logistic growth term in the autocatalysis term. These results are summarized in a new supplementary section:

Additional models with nonlinear mechanisms.

In the circuit scan described so far, we showed that saturation of the SnC removal mechanism is required to explain the longitudinal trajectories of Burd et al. The saturation effect is modeled using a Michaelis-Menten term for the SnC-dependent reduction in removal rate, $\frac{1}{1+\beta_2 X}$. This term introduces a non-linear dependence on SnC abundance, X . The autocatalysis effect, on the other hand, is modeled using a linear dependence on SnC abundance. To test whether other non-linear effects may explain the longitudinal trajectories, we tested several other plausible models. We kept the removal term linear in X , and added non-linearity to the production term.

First, we tested models where there is no saturation, and the non-linearity is introduced from logistic effects. In the first model, the logistic effect is introduced in the total production of SnC,

$$\dot{X} = (\eta_0 + \eta_1 t)(1 + \eta_2 X)(1 - \eta_3 X) - (\beta_0 - \beta_1 t)X + \sqrt{2\epsilon}\xi_t$$

and in the second model, it is introduced specifically in the autocatalysis factor:

$$\dot{X} = (\eta_0 + \eta_1 t)(1 + \eta_2 X(1 - \eta_3 X)) - (\beta_0 - \beta_1 t)X + \sqrt{2\epsilon}\xi_t$$

Both models do not improve on the USR model (same equations with $\eta_3=0$). They have a log-likelihood of -500, and the best-fit values for η_0, η_1 are 0.

We also tested a model where autocatalysis has a quadratic term $\eta_3 X^2$:

$$\dot{X} = (\eta_0 + \eta_1 t)(1 + \eta_2 X + \eta_3 X^2) - (\beta_0 - \beta_1 t)X + \sqrt{2\epsilon}\xi_t$$

This model improves on the other USR models, and has a maximal log-likelihood of $L = -486$ (BIC=1012). This model is much worse than the best-fit SR model ($\Delta\text{BIC}=34$). We therefore conclude that these models are insufficient to explain the longitudinal trajectories.

2. In Fig. 1 it would be very helpful to label the arrows of the model schematic with their associated parameters.

We have now labeled the arrows in Fig. 1 as suggested.

[redacted]

3. In Fig. 2, at 80 weeks the variance in TBL is large, with many mice still exhibiting low levels. Is it by chance that the two trajectories shown in Fig. 2A do not lie close to the mean trajectory of the data? It would be helpful to plot more (all?) of these trajectories overlapping, to get a better sense of these. It seems that many mice do not show increases in TBL over their lifetimes. This needs further investigation,

as it may suggest that while the SR model fits the mean SnC dynamics, trajectories for individual mice may be fit better by alternative mechanisms?

We thank the reviewer for this comment. We addressed the trajectories in a new supplementary section, titled “Models with individual variation in parameters”. This section includes a plot of all the longitudinal trajectories of Burd et al. It also includes an analysis that compares the SR model to an alternative model in which accumulation of SnC is deterministic (no noise), with a different slope for each individual (deterministic individuality, DI, in which slopes are drawn from the observed slope distribution and each individual maintains the slope throughout the dynamics). The SR model captures the average rank much better than the DI model. The SR model shows a slope histogram that captures that of the data trajectories. We also note that the model fits used all of the trajectory data, and not only mean statistics. We believe that this section helps to clarify the nature of the longitudinal trajectories and the way that the SR model fits them.

Models with individual variation in parameters

Figure S1. Individual variation in SnC accumulation rate can be described by stochastic models. (A) Individual trajectories of SnC accumulation from Burd et al., as in Figure 1A. Each color denotes a different mouse. (B) Mean rank of every individual from Burd et al. (light green) compared with rank distributions from a model of Deterministic Individuality (DI, purple, model illustrated in inset) or the SR model (black). (C) Slope of SnC accumulation rate for every individual from Burd et al. (light green) compared with slope distribution predicted from the parametrized SR model.

So far, we modeled individual variation in SnC accumulation as resulting from stochasticity in X which we described using a white noise term $\sqrt{2\epsilon}\xi_t$. Variation in senescent cell accumulation may also result from inter-individual variation in model parameters. This inter-individual variation in parameters is relevant for the case of humans, who vary in genotype and environment. There is also likely to be some

variation in model parameters between the inbred mice, whose SnC levels were measured longitudinally by Burd et al (Figure S1A). To test whether it is justified to model variation in model parameters between individuals, we estimated the persistent variation between individuals. For this, we calculated the rank of every individual at each time point, and estimated the mean rank of every individual throughout their lifetime (Figure S1B). In the most extreme case of inter-individual variation, where individuals have separate trajectories of SnC accumulation throughout their lifetime, we expect this distribution to be uniform (every individual maintains its rank throughout its lifetime, a ‘deterministic individuality’ model). On the other hand, the stochastic variation described by the parametrized SR model suggests a much narrower variation in mean rank, which is similar to what is observed in the dataset.

Finally, we estimated the variation in SnC accumulation by calculating the slope of SnC accumulation for every individual (Figure S1C). This corresponds well with the variation in SnC accumulation rates described by the parametrized SR model (Figure S1C). We conclude that stochastic models are adequate for describing variation in SnC dynamics for the Burd et al. dataset.

We note that the comparison of models to the longitudinal data in the main text was done with the full data trajectories, and not to their average statistics.

4. Lines 181-186: hard to follow. Please expand on this section as it is unclear. Also, here (and elsewhere) it can be difficult to follow the references to the SI - please make these more specific wherever possible, e.g. to a specific SI figure rather than to a section.

We improved and expanded the text according to this suggestion. We defined temporal scaling, and added references to the appropriate supplementary figure panels.

We further tested whether the SR model can explain the survival curves of *C. elegans* under different life-extending genetic, environmental and diet perturbations. These perturbations change mean lifespan by up to an order of magnitude. The survival curves show a remarkable feature called temporal scaling: the survival curves collapse onto approximately the same curve when age is scaled by mean lifespan (Figure 4F insets). That is, the entire distribution of death times, including its mean and standard deviation, is determined by a single parameter, which depends on the perturbation. We find that the SR model provides the shape of the survival curves, as well as their temporal scaling feature. Temporal scaling is found in the SR model by assuming that the perturbations affect the accumulation rate η (Figure 4F, Supplementary Section 9 and Supplementary Figure S10A).

Temporal scaling cannot be explained by models without rapid turnover (Supplementary Figure S10B), or by varying any other parameter except η in the SR model. Thus, we predict loss of temporal scaling of survival curves when a perturbation affects other SR-model parameters such as removal rate β

or noise ϵ (Supplementary Figure S10CD). This prediction may apply to exceptional perturbations in which temporal scaling is not found, such as the *eat-2* and *nuo-6* mutations (Supplementary Figure S10EF). We conclude that the SR model of rapid turnover with critical-slowness down is a candidate explanation for the temporal scaling of survival curves in *C. elegans*.

5. *Supplementary Section 2 Eq. 3 needs further justification/clarification: should the term $\beta \kappa \log(\kappa + X)$ be $\beta X \log(\kappa + X)$? This has implications for the following quasi-steady state analysis.*

We have rechecked this calculation. The potential function is:

$$U(X) = (\beta - \eta t)X - \beta \kappa \log(\kappa + X)$$

Since:

$$-\frac{d}{dx}U(X, t) = -(\beta - \eta t) + \frac{\beta \kappa}{X + \kappa} = \eta t + \frac{-\beta(\kappa + X) + \beta \kappa}{X + \kappa} = \eta t - \frac{\beta X}{X + \kappa}$$

We now added this derivation to the Supplementary Section.

6. *Please clarify the meaning of the term “ExpIntegralE” in the SI.*

We now clarified the meaning of the term in the relevant SI section. We added the sentence:

Where ExpIntegralE describes the exponential integral function $\text{ExpIntegralE}[n, z] = \int_1^\infty \frac{e^{-zt}}{t^n} dt$.

In summary, the reviewer comments helped us improve the manuscript by adding analysis of additional primary data and mathematical models, and by improving the clarity and rigor of the paper.

REVIEWERS' COMMENTS:

Reviewer #1 (Remarks to the Author):

The reviewers have addressed all my concerns. I take this opportunity to congratulate them for this nice work.

Reviewer #2 (Remarks to the Author):

In the revision the authors have satisfactorily addressed all of my questions.

Reviewer #1 (Remarks to the Author):

The reviewers have addressed all my concerns. I take this opportunity to congratulate them for this nice work.

Thank you.

Reviewer #2 (Remarks to the Author):

In the revision the authors have satisfactorily addressed all of my questions.

Thank you.